Wearable accelerometers reveal objective assessment of walking symmetry and regularity in idiopathic scoliosis patients

Gan Xiaopeng
Liu Xin
Cai Danxian
Zhang Rongbin
Li Fanqiang
Fang Haohuang
Huang Jingrou
Qiu Chenguang
Zhan Hongrui zhanhongrui@mail.sysu.edu.cn
Department of Rehabilitation Medicine, The Fifth Affiliated Hospital of Sun Yat-sen University , Zhuhai , China
Barbosa Neto Octavio
Electronic publication date: 2024 Jul 16
Publication date: 2024
Volume: 12
Electronic Location ID: e17739
Received 2024 Apr 12; Accepted 2024 Jun 23
Copyright: ©2024 Gan et al.
Copyright year: 2024
Copyright holder: Gan et al.
License: This is an open access article distributed under the terms of the Creative Commons Attribution License, which permits unrestricted use, distribution, reproduction and adaptation in any medium and for any purpose provided that it is properly attributed. For attribution, the original author(s), title, publication source (PeerJ) and either DOI or URL of the article must be cited.
License URL: https://creativecommons.org/licenses/by/4.0/

Keywords: Scoliosis, Walking stability, Gait analysis, Wearable sensors

Funding: Fifth Affiliated Hospital of Sun Yat-sen University This work was supported by the Fifth Affiliated Hospital of Sun Yat-sen University. The funders had no role in study design, data collection and analysis, decision to publish, or preparation of the manuscript.

==============================
Background

Scoliosis is a multifaceted three-dimensional deformity that significantly affects patients’ balance function and walking process. While existing research primarily focuses on spatial and temporal parameters of walking and trunk/pelvic kinematics asymmetry, there remains controversy regarding the symmetry and regularity of bilateral lower limb gait. This study aims to investigate the symmetry and regularity of bilateral lower limb gait and examine the balance control strategy of the head during walking in patients with idiopathic scoliosis.

Methods

The study involved 17 patients with idiopathic scoliosis of Lenke 1 and Lenke 5 classifications, along with 17 healthy subjects for comparison. Three-dimensional accelerometers were attached to the head and L5 spinous process of each participant, and three-dimensional motion acceleration signals were collected during a 10-meter walking test. Analysis of the collected acceleration signals involved calculating five variables related to the symmetry and regularity of walking: root mean square (RMS) of the acceleration signal, harmonic ratio (HR), step regularity, stride regularity, and gait symmetry.

Results

Our analysis reveals that, during the walking process, the three-dimensional motion acceleration signals acquired from the lumbar region of patients diagnosed with idiopathic scoliosis exhibit noteworthy disparities in the RMS of the vertical axis (RMS-VT) and the HR of the vertical axis (HR-VT) when compared to the corresponding values in the healthy control (RMS-VT: 1.6 ± 0.41 vs. 3 ± 0.47, P < 0.05; HR-VT: 3 ± 0.72 vs. 3.9 ± 0.71, P < 0.05). Additionally, the motion acceleration signals of the head in three-dimensional space, including the RMS in the anterior-posterior and vertical axis, the HR-VT, and the values of step regularity in both anterior-posterior and vertical axis, as well as the values of stride regularity in all three axes, are all significantly lower than those in the healthy control group (P < 0.05).

Conclusion

The findings of the analysis suggest that the application of three-dimensional accelerometer sensors proves efficacious and convenient for scrutinizing the symmetry and regularity of walking in individuals with idiopathic scoliosis. Distinctive irregularities in gait symmetry and regularity manifest in patients with idiopathic scoliosis, particularly within the antero-posterior and vertical direction. Moreover, the dynamic balance control strategy of the head in three-dimensional space among patients with idiopathic scoliosis exhibits a relatively conservative nature when compared to healthy individuals.

Introduction

Scoliosis is a three-dimensional deformity involving the skeletal and joint systems, typically accompanied by vertebral rotation in the transverse plane, alterations in the physiological curvature of the spine in the sagittal plane, and lateral curvature deformities in the coronal plane (Labrom et al., 2021). The Scoliosis Research Society (SRS) defines scoliosis as the measurement of the Cobb angle in the coronal plane X-ray imaging of the entire spine. If the Cobb angle is greater than 10 degrees and there is evident vertebral rotation, it is diagnosed as scoliosis (Negrini et al., 2018). Idiopathic scoliosis is a complex, multifactorial, and etiologically unclear neuro-musculo-skeletal disorder (Trobisch, Suess & Schwab, 2010).

Previous studies have indicated that patients with scoliosis experience impaired balance function, with asymmetry in the spine of scoliotic individuals not only altering the position of the center of mass in the trunk but also affecting the patient’s balance control strategy (Daryabor et al., 2017). Even in cases of mild spinal deformity (Cobb angle less than 25°), individuals with scoliosis demonstrate certain balance control impairments in simple postural control tasks (Haumont et al., 2011). As the severity of spinal deformity increases, the impairment in balance control becomes more pronounced, and individuals tend to adopt a more conservative method of balance control (Wu et al., 2020). Current research attributes the insufficient postural balance control in individuals with scoliosis to two hypotheses: the biomechanical hypothesis and the sensory integration hypothesis (Dufvenberg et al., 2018). The former suggests that spatial displacement of the spinal deformity and the positions of the shoulders, pelvis, and head contribute to the decline in postural control. The latter proposes that inaccurate sensory input weights lead to compromised dynamic regulation of sensorimotor integration, causing a decline in balance control abilities.

Impairments in postural balance control capacity caused by idiopathic scoliosis directly influence the spatiotemporal parameters of patients’ gait, resulting in abnormal gait patterns. Studies indicate that, compared to healthy subjects, individuals with idiopathic scoliosis exhibit a shorter stance phase and a longer swing phase (Liu et al., 2022). Correspondingly, pelvic joint activity and hip joint activity in the horizontal and sagittal planes during walking are reduced (Daryabor et al., 2017). However, previous research has primarily focused on the kinematics of the trunk and pelvis in scoliosis patients and changes in gait spatiotemporal parameters. Limited research exists on the symmetry of bilateral gait, and there is no consensus on the symmetry and regularity of bilateral lower limb gait (Daryabor et al., 2017). Moreover, most previous studies have analyzed gait spatiotemporal parameters and the motion of the trunk and pelvis to determine gait symmetry in individuals with scoliosis, using complex equipment such as plantar pressure plates or infrared three-dimensional motion capture systems (Sklensky et al., 2022; Wu et al., 2020; Yang et al., 2013; Zhu et al., 2021). However, direct measurements of gait symmetry and regularity were not commonly performed.

Spinal curvature in the sagittal, coronal, and horizontal planes alters the balance stability of patients in three-dimensional space. Neurosensory regulatory function maintains the trunk’s static and dynamic stable upright posture, while abnormal head positioning affects both the head and trunk posture, ultimately leading to decreased trunk stability (Beaulieu et al., 2009; Herman et al., 1985; Wen et al., 2022). Furthermore, maintaining head stability is essential for optimizing visual conditions, and maintaining head stability may be one of the fundamental tasks of the body posture control system during walking (Bril & Ledebt, 1998; Cromwell, Newton & Carlton, 2001).

Studies have shown that, based on imaging measurements and pressure analysis of the plantar surface, patients with scoliosis and spinal imbalance exhibit greater head offset distances than scoliosis patients in a balanced group. Additionally, the percentage difference in load between left and right feet and between front and rear feet is greater than in the balanced group. The increased instability of the head and body during static standing exacerbates trunk imbalance (Wen et al., 2022). However, there is currently a lack of research on the three-dimensional dynamic balance control strategies of the head in scoliosis patients during walking and whether they affect the symmetry and regularity of bilateral walking.

Gait symmetry and regularity are paramount in evaluating the walking stability and postural control of patients with idiopathic scoliosis. Conventional assessment techniques, which often employ expensive equipment like three-dimensional motion capture systems and force plates (Ji et al., 2024; Sklensky et al., 2022; Zhu et al., 2021), are confined to laboratory settings, presenting time-consuming and limited real-world applicability. In contrast, three-dimensional accelerometers, with their inherent portability, low cost, and suitability for long-term use, facilitate real-time gait monitoring. They are a preferred clinical instrument for assessing the gait of scoliosis patients, offering a practical and efficient alternative to sophisticated devices such as three-dimensional motion capture systems and force plates.

Therefore, this study aims to comprehensively investigate the symmetry and regularity of bilateral lower limb gait in patients with scoliosis using data collected through three-dimensional accelerometers. Simultaneously, the study investigates the balance control strategies of the head during the walking process. Through this research, we hope to provide a convenient measurement method for assessing the abnormal gait in patients with scoliosis, and simultaneously offer new insights for future clinical practices.

Materials & Methods

Ethics clearance

This study was conducted in accordance with the principles of the Helsinki Declaration. The research protocol received approval from the Medical Ethics Committee of the Fifth Affiliated Hospital of Sun Yat-sen University (Approval Number: Sun Yat-sen University Fifth Hospital [2022] Ethics No. (K34-1). The trial has been registered with the Chinese Clinical Trial Registry (Registration Number: ChiCTR2200061676). Informed consent for participation in the study was obtained from all participants themselves or their legal guardians.

Participant

This retrospective, single-center study was approved by our institutional ethics committee, and all subjects signed informed consent forms to participate. Idiopathic scoliosis subjects were recruited from December 2022 to June 2023. Inclusion criteria were as follows: (1) Diagnosis by a rehabilitation doctor as Lenke type 1 (main thoracic curve) or Lenke type 5 (main lumbar curve) idiopathic scoliosis (Lenke, 2005); (2) age greater than 10 years; (3) inclusion of idiopathic scoliosis patients with complete standing anteroposterior and lateral full-spine X-rays, with Cobb angles of the scoliotic curve greater than 10 degrees; (4) no history of wearing orthosis or surgery, no physical therapy or exercise training in the three months prior to inclusion; (5) bilateral lower limb length difference within a range of one cm, capable of independent standing and walking; (6) dominant foot on the left side; (7) absence of flat feet. Exclusion criteria were: (1) Diagnosis of non-idiopathic scoliosis; (2) severe deformities in the lower limbs and feet or deformities in other body parts; (3) presence of other diseases affecting gait, such as trauma, muscle atrophy, or joint diseases related to musculoskeletal disorders. This study recruited subjects with normal spinal alignment and aged over 10 years as the control group through posters and online advertisements in the hospital.

Data collection

The experiment employed wearable three-dimensional acceleration sensors (Xsens DOT, Enschede, NL) (weight = 11.2 g, dimensions = 36.3 × 30.4 × 10.8 mm) to collect three-dimensional plane (sagittal, coronal, horizontal) motion acceleration signals separately at the head and lumbar. The acceleration sensor had an internal sampling rate of 800 Hz, an output rate of 60 Hz, and a standard full range of ±16 g. Dynamic mode was utilized for data collection, and the output acceleration signals were gravity-acceleration-removed. The lumbar acceleration sensor was placed near the center of mass at the L5 spinous process using medical-grade double-sided tape and elastic bandage for fixation. The head sensor was positioned at the occipital pole using a headgear elastic band and medical-grade double-sided tape for fixation. The X-axis of the sensor pointed in the anatomical antero-posterior direction (AP), the Y-axis in the medial-lateral direction (ML), and the Z-axis in the vertical direction relative to the ground (VT).

10-meter walking test

During the 10-meter walking test, motion acceleration signals from the head and lumbar of the subjects were simultaneously collected. Subjects were instructed not to turn their heads, keep their eyes straight ahead, and let their arms swing naturally. Upon receiving the start command, subjects walked forward in a natural state for 15 m. Acceleration data for the 15-meter walk and the time taken by the subjects to traverse the central 10 m were recorded. The test was repeated twice, and the mean of the two results was considered as the final measurement outcome.

Data processing

The acceleration signals from the subjects were kept blinded to the data analysis researchers, and the data results were grouped only after the completion of the entire data analysis. The raw acceleration data, from which the gravity component had been removed, underwent filtering and smoothing using MATLAB (R2014a; the MathWorks, Inc., Natick, MA, USA). A second-order Butterworth low-pass filter with a cutoff frequency of 10 Hz was applied to ensure the representativeness and statistical significance of the results. Specifically, 420 points (7 s) from the middle of the acceleration signal were randomly selected as the data for calculating variables (Fig. 1). Five variables reflecting walking symmetry and regularity were selected and calculated in MATLAB.

Figure 1 Raw accelerometer data from the lumbar of a healthy participant during a 10-meter walking test.

A 420-point segment (7 s) was randomly extracted from the middle of the acceleration signal for analysis and computation. AP, antero-posterior direction; ML, medial-lateral direction; VT, vertical direction.

Walking symmetry and regularity variables

The variables for each walking symmetry and regularity parameter are derived from calculations based on the X (AP), Y (ML), and Z (VT) axis data of the acceleration signal. The specific parameters include:

Acceleration root mean square (RMS): RMS measures the dispersion of the measured acceleration signal relative to zero. It’s a commonly used parameter to evaluate the dynamic stability of measured segments during walking, reflecting the fluctuation of measured segments during walking (Menz, Lord & Fitzpatrick, 2003).

Harmonic ratio (HR): The HR represents the smoothness and rhythmicity of the acceleration pattern. The HR on the anterior-posterior axis (X) and the sagittal axis (Z) is calculated as the ratio of the sum of even harmonic values to the sum of odd harmonic values. While for the acceleration on the medial-lateral axis (Y), the HR is calculated as the ratio of the sum of odd harmonic values to the sum of even harmonic values, contributing to the assessment of acceleration in the medio-lateral direction (Menz, Lord & Fitzpatrick, 2003).

Step regularity (AD1): AD1 represents the amplitude of the first peak in the acceleration autocorrelation signal.

Stride regularity (AD2): AD2 represents the amplitude of the second peak in the acceleration autocorrelation signal.

Higher AD1 and AD2 values closer to 1 indicate better step regularity and stride regularity, respectively (Moe-Nilssen & Helbostad, 2004) (Fig. 2).

Figure 2 Example of unbiased autocorrelation coefficient sequence derived from head accelerometry time series captured during gait in a healthy individual (orange-yellow curve) and a patient with idiopathic scoliosis (blue dotted curve).

Output has a length of 600 samples. D1 denotes the phase lag for one step, and D2 corresponds to the phase lag for one complete stride. The magnitude AD1 and AD2 represents step regularity and stride regularity, respectively. The graph reveals that the healthy participant’s magnitudes AD1 and AD2 are greater than those of the idiopathic scoliosis patients, indicating a higher level of regularity. AP, antero-posterior direction, ML, medial-lateral direction, VT, vertical direction.

Gait symmetry: Gait symmetry is reflected by the closeness of AD1/AD2 to 1.0. A value closer to 1.0 indicates greater symmetry in gait (Moe-Nilssen & Helbostad, 2004).

These parameters are indicative of various aspects of gait stability, smoothness, regularity, and symmetry and are calculated based on different aspects of the acceleration signal across multiple axes.

Walking speed and cadence

The formula for calculating walking speed (V) is given by V =10/t, where t represents the time taken to walk 10 m. The formula for calculating cadence (c) is c =60f/N, where f is the sampling frequency (60 Hz in this study), N is the number of samples per dominant period, and c is the cadence expressed in steps per minute (steps/min) (Menz, Lord & Fitzpatrick, 2003).

Statistical analysis

Statistical analysis of experimental data was performed using SPSS 20.0 (IBM, Chicago, IL, USA). Measurement results were expressed as mean ± standard deviation for each group. Initially, the Kolmogorov–Smirnov normality test was conducted. For data conforming to normal distribution, an independent t-test was employed to analyze the differences in walking symmetry and regularity variables between the two groups of subjects. For data that did not pass the normality test, the Mann–Whitney U test was applied as a non-parametric test. A significance level of P < 0.05 was considered statistically significant.

Results

Participant characteristics

This study included a total of 17 patients with Lenke 1 and Lenke 5 types of idiopathic scoliosis, as well as 17 healthy subjects. The demographic characteristics of the two groups, including age, weight, and gender ratio, were similar (p > 0.05) (Table 1). Among the 17 included patients with idiopathic scoliosis, six were classified as Lenke 1 type, and 11 as Lenke 5 type. Each idiopathic scoliosis subject provided complete standing anteroposterior and lateral full-spine X-rays, with an average Cobb angle of 21.75 ± 6.56 for the main curve.

Table 1 Mean value and standard deviation of subject characteristics.

Parameters	IS (n = 17)	Healthy (n = 17)	P-value	
Age (year)	22.59 + 5.7	22.47 ± 3.78	0.94	
Weight (Kg)	49.85 + 6.02	55 ± 9.11	0.06	
Height (cm)	159.47 + 9.54	162.65 ± 7.34	0.29	
BMI (kg/m2)	19.58 + 1.29	20.7 ± 2.34	0.1	
Notes.

BMI body mass index

IS idiopathic scoliosis

Walking symmetry and regularity variables

The analysis results of walking symmetry and regularity variables show significant changes in the measurements of the lumbar and head regions for subjects with idiopathic scoliosis compared to healthy subjects (Table 2).

Table 2 Walking symmetry and regularity variables between idiopathic scoliosis patients and healthy subjects.

Variables	IS	Healthy	
Walking speed(m/s)	1.28 ±0.07	1.29 ±0.08	
Cadence(steps/min)	115.5 ±6.51	117.97 ±5.7	
RMS			
Head			
AP	0.49 ± 0.22*	0.65 ± 0.22	
ML	0.76 ±0.24	0.84 ±0.21	
VT	2.02 ± 0.4*	2.45 ± 0.49	
Lumber			
AP	1.83 ±0.29	1.96 ±0.23	
ML	1.35 ±0.35	1.51 ±0.42	
VT	1.96 ± 0.41*	2.43 ± 0.47	
HR			
Head			
AP	1.39 ±0.51	1.59 ±0.57	
ML	2.98 ±0.72	3.14 ±0.69	
VT	3.37 ± 0.83*	4.35 ± 0.84	
Lumber			
AP	3.72 ±0.87	3.52 ±0.99	
ML	2.02 ±0.44	2.22 ±0.67	
VT	3.45 ± 0.72*	3.97 ± 0.71	
Step regularity			
Head			
AP	0.39 ± 0.28*	0.58 ± 0.11	
ML	−0.74 ±0.15	−0.75 ±0.13	
VT	0.88 ± 0.06*	0.93 ± 0.04	
Lumber			
AP	0.87 ±0.07	0.84 ±0.13	
ML	−0.5 ±0.15	−0.57 ±0.19	
VT	0.87 ±0.06	0.9 ±0.07	
Stride regularity			
Head			
AP	0.65 ± 0.13*	0.79 ± 0.1	
ML	0.82 ± 0.07*	0.87 ± 0.05	
VT	0.89 ± 0.06*	0.94 ± 0.03	
Lumber			
AP	0.89 ±0.06	0.88 ±0.09	
ML	0.69 ±0.11	0.76 ±0.11	
VT	0.87 ±0.06	0.9 ±0.06	
Gait symmetry			
Head			
AP	0.58 ±0.4	0.75 ±0.15	
ML	−0.89 ±0.14	−0.87 ±0.16	
VT	0.98 ±0.06	0.99 ±0.02	
Lumber			
AP	0.98 ±0.05	0.96 ±0.13	
ML	−0.73 ±0.22	−0.8 ±0.28	
VT	0.998 ±0.03	0.996 ±0.03	
Notes.

IS idiopathic scoliosis

RMS root mean square

HR harmonic ratio

AP antero-posterior direction

ML medial-lateral direction

VT vertical direction

* p < 0.05.

Bold indicates statistically significant values.

In comparison to healthy individuals, patients with idiopathic scoliosis exhibit a decrease in RMS in both the lumbar and head regions across all three directions, but only the RMS in the VT exhibits a significant difference (p < 0.05). For the RMS on the ML, there is no significant difference in the head and lumbar regions between the idiopathic scoliosis group and the healthy control group (p > 0.05). In the AP direction, although the head RMS of the idiopathic scoliosis group is significantly lower than that of the healthy control group (0.49 ± 0.22 vs. 0.65 ± 0.22 respectively, p < 0.05), there is no significant difference between the two groups in the lumbar region (p > 0.05) (Fig. 3). In regard to the HR, the overall HR values of idiopathic scoliosis subjects are smaller than those of the healthy control group, but statistical differences are only present in the VT direction (p < 0.05) (Fig. 4).

Figure 3 Differences in root mean square (RMS) (*p < 0.05).

S, idiopathic scoliosis group; C, healthy control group; AP, antero-posterior direction; ML, medial-lateral direction, VT, vertical direction.

Figure 4 Differences in harmonic ratio (HR) (*p < 0.05).

S, idiopathic scoliosis group; C, healthy control group; AP, antero-posterior direction; ML, medial-lateral direction, VT, vertical direction.

When evaluating the regularity of gait, the step regularity of the head in idiopathic scoliosis significantly decreases in both the AP and VT directions (p < 0.05) (Fig. 5). Additionally, the stride regularity in all three directions of the head is significantly lower than that of the healthy control group (p < 0.05) (Fig. 6). However, for the lumbar region, both step regularity and stride regularity in all three directions show no significant difference compared to the healthy control group (p > 0.05).

Figure 5 Differences in step regularity (*p < 0.05).

C, healthy control group; AP, antero-posterior direction; ML, medial-lateral direction; VT, vertical direction.

Figure 6 Differences in stride regularity (*p < 0.05).

S, idiopathic scoliosis group; C, healthy control group; AP, antero-posterior direction; ML, medial-lateral direction; VT, vertical direction.

The gait symmetry in the head and lumbar regions between the idiopathic scoliosis group and the healthy control group does not exhibit significant differences across all three directions (p > 0.05). Additionally, no significant distinctions were found in terms of 10-meter walking speed and cadence when comparing the two groups (p > 0.05) (Table 2).

Discussion

To the best of our knowledge, this is the first study employing wearable three-dimensional accelerometer sensors to analyze the gait regularity and symmetry of individuals with idiopathic scoliosis. The results indicate significant issues in both gait regularity and symmetry for patients with Lenk1 and Lenk5 types of idiopathic scoliosis, particularly evident in the analysis of acceleration signals collected from the head.

RMS values represent the average amplitude of acceleration sensors in three-dimensional directions during walking tests. The analysis conducted in this study on the walking regularity and symmetry among subjects with idiopathic scoliosis reveals that the RMS values recorded in the head and lumbar are lower than those observed in the healthy control group. Notably, there are significant differences in the RMS values of the VT direction in the head and lumbar, as well as the AP direction in the head (p < 0.05), when compared to the healthy control group. However, no significant difference is found in the RMS values in the ML direction and the three directions in the lumbar (p > 0.05). In previous studies, an increase in RMS was considered an indication of decreased walking balance control and a manifestation of walking instability (Osaka et al., 2017; Zhang et al., 2021). However, in this study, with no significant difference in the 10 m walking speed between the two groups, RMS values for idiopathic scoliosis decreased. This finding might be attributed to a specific conservative balance control strategy adopted by individuals with idiopathic scoliosis. As previously suggested, RMS in the VT direction reflects the displacement of the center of gravity (COG) during walking (Osaka et al., 2017). The study aligns with previous research, indicating that individuals with idiopathic scoliosis exhibit reduced displacement in the VT direction of the head and lumbar during walking compared to healthy individuals, implying a more cautious walking pattern (Mahaudens et al., 2009). Mahaudens et al. (2018) proposed that this gait pattern in individuals with idiopathic scoliosis represents an economical way to avoid excessive metabolic energy consumption. Consequently, our study suggests that individuals with idiopathic scoliosis employ conservative and cautious balance strategies and maintain a stable visual field in the VT and AP direction during the walking process to avoid gait imbalance.

In this study, the HR values in the VT direction of the head and lumbar region in patients with idiopathic scoliosis are significantly lower than those of healthy subjects. HR is utilized as an indicator to evaluate step-to-step symmetry within a stride, where a higher HR value indicates stronger symmetry between gaits and a more stable walking pattern (Bellanca et al., 2013). Previous research has employed acceleration data from the lumbar and head to compute HR and evaluate gait stability among patients with hemiplegia, the elderly, parkinson’s patients, and individuals with dizziness (Castiglia et al., 2022; Doi et al., 2013; Mitsutake et al., 2022; Zhang et al., 2021). Findings have demonstrated that as the disparity in limb loading conditions increases, the kinematics and dynamics of both lower limbs become more asymmetric during walking, resulting in a smaller HR value (Bellanca et al., 2013). Previous studies have indicated a significant asymmetry in the loading of the lower limbs on both sides during walking in patients with idiopathic scoliosis (Park et al., 2016; Sklensky et al., 2022), with an abnormal lower limb loading pattern ultimately leading to decreased balance (Márkus et al., 2018). In summary, this study suggests that patients with idiopathic scoliosis exhibit step-to-step asymmetry within a stride during walking.

On the other hand, this study also examines changes in walking stability in patients with idiopathic scoliosis from the perspective of gait quality. The step regularity and stride regularity of idiopathic scoliosis are significantly lower than those of healthy subjects, suggesting that patients with idiopathic scoliosis have poorer regulation of repetitive walking speed and control over body rhythm displacement during walking. However, this disparity is observed only in the acceleration signals collected from the head, while the signals obtained from the lumbar exhibit less noticeable variation between the two groups. This discrepancy may be related to the deformities caused by scoliosis. Apart from altering the overall movement control strategy of the trunk (Wu et al., 2020; Wu et al., 2019; Yang et al., 2013), the spinal deformity also directly affects the stability control capability of the head (Kučerová et al., 2023; Wen et al., 2022). Consequently, during walking, with the head being the top of the trunk, its movement strategy is likely to be more significantly impacted by the spinal deformities.

Three-dimensional accelerometers are wearable devices that offer high practicality, low cost, and convenience. Previous studies have demonstrated their clinical application in assessing patients’ walking stability and fall risk, with high reliability (Gillain et al., 2018; Jarchi et al., 2018). This study not only highlights the differences in walking symmetry and regularity between patients with idiopathic scoliosis and healthy controls but also confirms the effectiveness of wearable three-dimensional accelerometers in evaluating these parameters and head balance control in idiopathic scoliosis patients. These findings underscore the extensive applicability of wearable sensors in future research and clinical settings for understanding motor control in idiopathic scoliosis, particularly in the early identification of gait abnormalities linked to the condition.

Limitations

The primary concern centers on the representativeness of the study sample. Exclusively comprising patients diagnosed with idiopathic scoliosis of Lenke 1 and 5, the sample entails a highly specific selection of research subjects. The average Cobb angle among individuals with scoliosis is a modest 21.75 ±  6.56, indicative of mild spinal curvature. This particularity in participant selection may potentially constrain the generalizability and representativeness of the research findings. A further consideration revolves around constraints associated with the employed technological apparatus. Despite the use of three-dimensional accelerometers as measurement instruments, this technology exhibits certain limitations. Specifically, accelerometers might fall short in furnishing intricate biomechanical insights, encompassing joint angles and muscle activity, potentially constraining a holistic comprehension of gait abnormalities. In subsequent investigations, it becomes imperative to systematically address these constraints, thereby augmenting the scientific rigor and trustworthiness of the study.

Conclusion

In summary, the investigation revealed that individuals with idiopathic scoliosis exhibit abnormalities in the symmetry and regularity of their gait during the walking process, discernible across the AP, ML, and VT direction. Furthermore, there is a relatively conservative dynamic balance control strategy for the head in three-dimensional space. The utilization of wearable three-dimensional accelerometers sensors presents a promising and convenient approach to assess the symmetry and regularity of walking in patients with idiopathic scoliosis. These research findings contribute novel insights into the gait symmetry, regularity, and measurement for individuals with idiopathic scoliosis, offering valuable references for prospective diagnostic. Subsequent studies should further investigate the correlation between lower limb gait asymmetry, spinal deformity progression, and relevant mechanical functional characteristics in patients with idiopathic scoliosis.

Supplemental Information

Supplemental Information 1 Acceleration data from the head during the 10-Meter Walking Test

Supplemental Information 2 Acceleration data from the lumbar during the 10-Meter Walking Test

Supplemental Information 3 Matlab code

Supplemental Information 4 STROBE checklist

We extend our sincere gratitude to all the participants and their families, as well as to the individuals who played a fundamental role in ensuring the successful completion of this study. Your active participation has been crucial in shaping the results, and we genuinely appreciate your valuable contributions to this research.

Additional Information and Declarations

Competing Interests

Author Contributions

Human Ethics

Clinical Trial Ethics

Data Availability

Clinical Trial Registration

The authors declare there are no competing interests.

Xiaopeng Gan conceived and designed the experiments, performed the experiments, authored or reviewed drafts of the article, and approved the final draft.

Xin Liu performed the experiments, prepared figures and/or tables, and approved the final draft.

Danxian Cai performed the experiments, prepared figures and/or tables, and approved the final draft.

Rongbin Zhang analyzed the data, prepared figures and/or tables, and approved the final draft.

Fanqiang Li conceived and designed the experiments, prepared figures and/or tables, and approved the final draft.

Haohuang Fang analyzed the data, prepared figures and/or tables, and approved the final draft.

Jingrou Huang performed the experiments, prepared figures and/or tables, and approved the final draft.

Chenguang Qiu analyzed the data, prepared figures and/or tables, and approved the final draft.

Hongrui Zhan conceived and designed the experiments, authored or reviewed drafts of the article, and approved the final draft.

The following information was supplied relating to ethical approvals (i.e., approving body and any reference numbers):

The Fifth Affiliated Hospital of Sun Yat-sen University

The following information was supplied relating to ethical approvals (i.e., approving body and any reference numbers):

The Fifth Affiliated Hospital of Sun Yat-sen University(Ethical Application Ref: Zhongda Wuyuan [2022] No. K34-1).

The following information was supplied regarding data availability:

The acceleration data from the lumbar and head during the 10-Meter Walking Test are available in the Supplementary Files.

The following information was supplied regarding Clinical Trial registration:

ChiCTR2200061676

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
