# Peer review of "Wearable accelerometers reveal objective assessment of walking symmetry and regularity in idiopathic scoliosis patients"

_PeerJ, doi:10.7717/peerj.17739_

## Round 0.1 · original submission · Minor Revisions

The study entitled “Wearable accelerometers reveal objective assessment of walking symmetry and regularity in idiopathic scoliosis patients” demonstrated excellent findings using an appropriate methodological approach. However, some points must be clarified in the manuscript. Your article has great potential for publication on PeerJ, but the reviewers have requested changes to be made.

Reviewer 1 ·

Basic reporting

The manuscript is well written and relevant for the field. The authors have cited important references and provided good contextualization. The results are aligned with the study hypothesis and the discussion is clear and objective.

Overall, I only have two comments regarding the Introduction and Discussion.

1) In the introduction, the authors do a good job with describing issues related with walking simmetry and regularity; however, the problematization regarding the importance of using wearable sensors for assessing such issues is not clear and deserves attention.

2) The authors should discuss further the importance, implications, and applications of the study results in the discussion. In the current state, the discussion majorly focus on the differences in simmetry and regularity in walking detected between the two groups. I believe the authors should also discuss the applications of the results in face of the growing use of wearable sensors and how the results advance the field forward.

Experimental design

The experimental design was appropriate and in line with the PeerJ criteria for acceptance.

Validity of the findings

The findings are valid and robust. Considering the methods and the careful conduction of the study, I have not concerns related to this item.

Additional comments

The study is of major importance in the field and will contribute to the specialized literature on the research theme.

·

Basic reporting

The proposed work studies the use of wearable sensor to monitor the symmetry and regularity of walking in individuals with idiopathic scoliosis. The authors demonstrate that different metrics based on accelerometric data coming from the lumbar and the head can be used to compare healthy individuals and individuals with idiopathic scoliosis
The article language is clear and easy to read and follow. I have concerns about the use of the word “lumber” throughout the paper; is it “lumbar” the right term? The authors should check this and other words, including some typos like missing spaces (e.g., line 67).
The article is well-structured, and figures and tables are descriptive; however, raw data coming from the accelerometers is not shared but only the results coming from the post-processing of the data. Also, please check the labels of the figures: in addition to the "lumber" word, in figure 5 there is the y-label is called "strip regularity" while the figure description says "stride regularity"
The data showed on figure 1 are interesting, but they could be coupled with data coming from idiopatic patients for comparison. Moreover, a visualization of the raw accelerometric data could be of impact for visualizing the cyclic behaviour at least of the healthy gait.

Experimental design

In the “Data Processing” section, it is not specified how the gravity component has been removed. It is not clear if there are multiple filter or only one second-order butterworth; also, please justify the choice of the second order and the 10 Hz cut-of. In an open-data and reproducibility prospective, could be interesting to share the code or the code repository that provided the results discussed.
On line 200 it is not clear how the results are expressed: one descriptive value for each participant? Please elaborate.

Validity of the findings

no comment

---

## Round 0.2 · accepted · Accept

After reviewing the changes you made based on the reviewers' recommendations, I must say I am impressed with the excellent work you have done.

·

Basic reporting

The previous comments have been resolved.

Experimental design

Regarding the filtering of the gravity component, it could be possible to add a sentence that makes clear that this action is made by the sensor processor itself.

I understand the authors' perspective regarding making raw data public. However, if possible, in case and after publication, I would suggest sharing the raw data on public repositories that are also citable (e.g., Zotero).

Validity of the findings

no comment

Additional comments

The authors have worked on the manuscript and they also enriched the supplementary data with the code used to process the raw data. I believe that the manuscript is now clearer and more complete thanks to the authors' work.